# Sida Golden Mosaic Virus, an Emerging Pathogen of Snap Bean (*Phaseolus vulgaris* L.) in the Southeastern United States

**DOI:** 10.3390/v15020357

**Published:** 2023-01-26

**Authors:** Saurabh Gautam, James W. Buck, Bhabesh Dutta, Timothy Coolong, Tatiana Sanchez, Hugh A. Smith, Scott Adkins, Rajagopalbabu Srinivasan

**Affiliations:** 1Department of Entomology, University of Georgia, 1109 Experiment Street, Griffin, GA 30223, USA; 2Department of Plant Pathology, University of Georgia, 1109 Experiment Street, Griffin, GA 30223, USA; 3Department of Plant Pathology, University of Georgia, 3250 Rainwater Road, Tifton, GA 31793, USA; 4Department of Horticulture, University of Georgia, 1111 Miller Plant Sciences, Athens, GA 30602, USA; 5University of Florida, IFAS Extension, 22712 W. Newberry Road, Newberry, FL 32669, USA; 6Department of Entomology and Nematology, University of Florida, 14625 Co Rd 672, Wimauma, FL 33598, USA; 7USDA-ARS, U.S. Horticultural Research Laboratory, Fort Pierce, FL 34945, USA

**Keywords:** *Begomovirus*, host range, phylogenetics, symptoms, *Bemisia tabaci*, vector–virus interactions

## Abstract

Sida golden mosaic virus (SiGMV) was first detected from snap bean (*Phaseolus vulgaris* L.) in Florida in 2006 and recently in Georgia in 2018. Since 2018, it has caused significant economic losses to snap bean growers in Georgia. This study, using a SiGMV isolate field-collected from prickly sida (*Sida spinosa* L.), examined the putative host range, vector-mediated transmission, and SiGMV-modulated effects on host–vector interactions. In addition, this study analyzed the phylogenetic relationships of SiGMV with other begomoviruses reported from *Sida* spp. Host range studies confirmed that SiGMV can infect seasonal crops and perennial weed species such as snap bean, hollyhock (*Alcea rosea* L.)*,* marsh mallow *(Althaea officinalis* L.), okra (*Abelmoschus esculentus* (L.) Moench), country mallow (*Sida cordifolia* L.), prickly sida (*S. spinosa*), and tobacco (*Nicotiana tabacum* L.). The incidence of infection ranged from 70 to 100%. SiGMV-induced symptoms and virus accumulation varied between hosts. The vector, *Bemisia tabaci* Gennadius, was able to complete its life cycle on all plant species, irrespective of SiGMV infection status. However, SiGMV infection in prickly sida and country mallow positively increased the fitness of whiteflies, whereas SiGMV infection in okra negatively influenced whitefly fitness. Whiteflies efficiently back-transmitted SiGMV from infected prickly sida, hollyhock, marsh mallow, and okra to snap bean, and the incidence of infection ranged from 27 to 80%. Complete DNA-A sequence from this study shared 97% identity with SiGMV sequences reported from Florida and it was determined to be closely related with sida viruses reported from the New World. These results suggest that SiGMV, a New World begomovirus, has a broad host range that would allow its establishment in the farmscapes/landscapes of the southeastern United States and is an emerging threat to snap bean and possibly other crops.

## 1. Introduction

Epidemics of emergent plant viruses are driven by a complex of interactions between ecological factors [1]. Exact causes favoring the emergence of vector-transmitted plant viruses are difficult to pin-point. However, availability of susceptible plants, congenial environmental conditions, and efficient vectors are minimum requirements to initiate epidemics [1,2,3]. In addition, intrinsic genetic traits that determine virus fitness in different host plants can have significant effects on epidemics [4]. Insect vectors, particularly invasive species, can function as significant drivers of plant virus emergence, and the vector host range could subsequently affect the plant host range of a virus [3]. The invasive sweet potato whitefly, *Bemisia tabaci* (Gennadius), has been responsible for emergence of several plant viruses in southeastern United States, particularly in the genus *Begomovirus* [5]. The broad host range of *B. tabaci* B cryptic species facilitates the epidemics of many begomoviruses in multiple hosts [6].

*Begomovirus* is the largest genus within the family *Geminiviridae* [7]. The viruses in this genus cause devastating diseases in multiple crops around the world [7]. These viruses, depending on their genome organization, are either monopartite or bipartite [8]. They exhibit distinct phylogeographical distribution, with most bipartite viruses being reported from the New World (NW), and most monopartite viruses being reported from the Old World (OW) [9,10]. Management of begomoviruses is a worldwide challenge, particularly in tropical and subtropical regions where both susceptible hosts and vectors are available year-round [11,12,13,14]. Within these regions, rapid evolution of begomoviruses has resulted in the establishment of numerous species that infect a given or related plant species. For instance, more than 30 and 90 begomovirus species are reported to infect fanpetals (*Sida* spp., family *Malvaceae*) and tomato (*Solanum lycopersicum* L., family *Solanaceae*), respectively [11]. Change in a single epidemiological factor such as presence of an efficient vector or a susceptible plant can transform sporadic incidences into reoccurring epidemics [15]. Therefore, the precise identification of begomoviruses, along with understanding the ecological factors such as host range, virus fitness in host plants, transmission efficiency by vectors, and genetic relationships with closely related viruses are crucial to developing effective management strategies.

Begomoviruses are transmitted exclusively by *Bemisia tabaci* (Gennadius) (Hemiptera: Aleyrodidae) cryptic species complex [11,16,17,18]. The mode of transmission is circulative and persistent, i.e., after acquisition, the virus circulates through the whitefly body before it can be inoculated, and post-acquisition, begomoviruses can remain associated for the rest of the insect’s lifespan [19]. Whether begomovirus can move transovarially from mother to offspring is not clear, with studies reporting both positive and negative outcomes [20,21,22,23]. Nevertheless, begomoviruses have become an emerging and serious threat to the production of many vegetable crops worldwide. One of the major factors behind the emergence of begomoviruses is the global spread of two cryptic species of *B. tabaci* viz., B and Q [12,24]. *Bemisia tabaci* B was first reported in the mid-1980s in the southeastern United States and has since become the predominant cryptic species in the region [25,26,27,28]. *Bemisia tabaci* B is highly polyphagous and is an efficient vector of several important viruses infecting vegetable crops in the southeastern United States [5,29,30].

In August 2018, snap bean, *Phaseolus vulgaris* L. (family *Fabaceae*) plants with characteristic begomovirus infection symptoms (crumpled, curled, and thickened leaves) were found in Tifton, Georgia, USA, and these snap bean plants were very heavily infested with whiteflies (Figure 1A). The plants were initially presumed to be infected with cucurbit leaf crumple virus (CuLCrV), but the virus inducing these symptoms was later identified as sida golden mosaic virus (SiGMV). SiGMV is a bipartite *Begomovirus* and was previously reported from *P. vulgaris* in Alachua County in Florida in 2006 [31]. In Georgia, since its first report in 2018, SiGMV has caused serious economic losses to snap bean growers [29]. SiGMV often occurs as a mixed infection with another whitefly transmitted begomovirus, CuLCrV, that causes the leaf crumple disease in snap bean and squash. Snap bean is an important vegetable crop in the southeastern United States, especially in Georgia, where in 2019 alone growers planted ~9873 acres of snap bean with a farmgate value in excess of USD 25 million [32].

The objectives of this study were to evaluate the prospective host range of SiGMV in the farmscapes of Georgia and Florida and investigate the ability of *B. tabaci* B to acquire and transmit SiGMV from susceptible hosts to snap bean. This study also evaluated the ability of these hosts to serve as vector reservoirs, and if virus infection in these hosts differentially modulated vector fitness that could favor epidemics of the virus. In addition, using complete DNA-A sequences, phylogenetic analysis was used to establish relationships of the SiGMV isolate used in the current study with other begomoviruses reported from *Sida* spp. in other parts of the world, including more recent isolates from Florida.

## 2. Materials and Methods

### 2.1. SiGMV Idenfication and Inoculum Source

#### 2.1.1. SiGMV Identification in Georgia

Twenty snap bean plant samples exhibiting begomovirus-associated symptoms were collected. Total DNA was extracted from 100 mg of leaf tissue of the symptomatic plants using the GeneJET Plant Genomic DNA Purification Kit (ThermoFisher Scientific, Waltham, MA, USA) following guidelines supplied by the manufacturer. Initial testing for the suspected CuLCrV using the protocols described earlier by Gautam et al. 2020 was negative [33]. Presence of another possible begomovirus in the samples was examined through PCR using the degenerate primer pair PAR1c496 and PAL1v1978 (5′-GCCCACATYGTCTTYCCNGT-3′ and 5′-GGCTTYCTRTACATRGG-3′), which amplified a 1.159 kb region of DNA-A of begomoviruses [34]. Gel analysis of PCR products confirmed the presence of begomovirus in all samples. PCR products from five of twenty samples were purified using the GeneJET PCR Purification Kit (ThermoFisher Scientific, Waltham, MA, USA) following instructions supplied by the manufacturer. Purified PCR products were sequenced in both directions and aligned to generate complete sequence using Geneious software [35]. Subsequent BLAST analysis of all five sequences confirmed the presence of a previously unreported begomovirus from Georgia. The sequences shared an up to 97% similarity with the DNA-A of SiGMV reported from Florida [31].

To further verify the SiGMV presence in the samples, SiGMV-specific primers were designed using Primer3 software [36] (Appendix A). PCR was performed for all twenty samples using primers SiGMVF and SiGMVR (Appendix A) that amplified a 574 bp region of AV1 gene (coat protein, CP) of SiGMV. For PCR, GoTaq Green Mastermix buffer (2X) (Promega, Madison, WI, USA) was combined with forward and reverse primers (SiGMVF and SiGMVR, Appendix A) (final concentration of 0.5 μM), 20 ng of DNA, and sterile nuclease-free water for a final volume of 10 μL. PCR was performed using a T-100 thermocycler (Bio-Rad, Hercules, CA, USA). An initial denaturation step (2 min at 95 °C) was followed by 40 cycles of 95 °C for 1 min, 60 °C for 1 min, and 72 °C for 1 min, ending with a final extension of 72 °C for 10 min. Gel analysis revealed the presence of the target amplicon in all samples. PCR products from five samples were sequenced, and subsequent BLAST analysis confirmed the presence of SiGMV in snap bean samples.

#### 2.1.2. SiGMV Identification in Florida

In October 2019, snap bean samples exhibiting begomovirus-like symptoms were collected from Alachua County, Florida. In November 2019, similar symptoms were twice observed in snap bean in St. Lucie County, Florida. Degenerate begomovirus primers described above and from Wyatt and Brown were used for PCR [34,37]. Subsequent sequence analysis of cloned amplicons (GenBank accession no. OQ266399-403) was used for begomovirus identification.

#### 2.1.3. SiGMV Inoculum Source for Greenhouse Experiments

In September 2018, prickly sida plants exhibiting putative golden mosaic symptoms were collected from just outside a *P. vulgaris* field in Tifton, Georgia (Figure 1B). Twenty plants were dug with a shovel and placed individually in plastic pots (2.37 L). The field-collected prickly sida plants were maintained in a greenhouse (25–30 °C, 14 h L:10 h D) in whitefly-proof cages (45 L × 45 W × 90 H) (Megaview Science Co., Taichung, Taiwan) at four plants per cage. Total DNA was extracted from 100 mg of symptomatic leaf tissue using the GeneJET Plant Genomic DNA Purification Kit and subjected to PCR analysis using the degenerate and SiGMVF and SiGMVR primer pairs as described above. Gel electrophoresis revealed the presence of target amplicons in four out of twenty samples. PCR products from four samples were sequenced to confirm the presence of SiGMV. Throughout the study, SiGMV was maintained in seed-germinated prickly sida plants through repeated inoculations with viruliferous whiteflies [38,39]. Prickly sida was used as an inoculum source for SiGMV instead of snap bean, as it was easier to maintain SiGMV in prickly sida plants.

### 2.2. Plants for Host Range Evaluation

Eight plant species representing three different families commonly present in the southeastern United States were selected. Seeds of prickly sida (*S. spinosa* L.) were collected from matured flowers of plants in the greenhouse. Mallow seeds were procured from different suppliers: country mallow (*Sida cordifolia* L.) from Asklepios Seeds (Bad Liebenzell, Germany); marsh mallow (*Malva parviflora* L.) from Outsidepride (Independence, OR); and hollyhock (*Alcea rosea* L.) from Outsidepride (Independence, OR, USA). Cotton (*Gossypium hirsutum* L. cv. ‘ST 6182 GLT’), okra (*Abelmoschus esculentus* L. cv. ‘Clemson Spineless 80′), and tobacco (*Nicotiana tabacum* L. cv. ‘L326′) seeds were obtained from the University of Georgia Extension Service (Tifton, GA, USA). Snap bean (*P. vulgaris* L. cv. Provider) seeds were obtained from Johnny’s selected seeds (Winslow, ME, USA). Plants were grown using Grower Mix (Asb Greenworld, West Point, VA, USA) in 10 cm plastic pots (depth 4 of cm). Two seeds per pot were sown and maintained in a greenhouse in whitefly-proof cages under the conditions described above. One week after germination, seedlings were thinned to one per pot. Plants at the two-true-leaf stage (~3–5 weeks old) were used for experiments.

### 2.3. Insects

The whiteflies (*B. tabaci* cryptic species B) used in the present study were first collected in Tifton, Georgia from cotton (*G. hirsutum* L.) in 2009 and have been reared on cotton plants since then in whitefly-proof cages in the greenhouse at above-stated conditions (GenBank Accession No. MN970031).

### 2.4. SiGMV Symptoms and Accumulation in Inoculated Host Plants

Viruliferous and non-viruliferous whiteflies were obtained by providing whiteflies with an acquisition access period (AAP) of 72 h on infected or non-infected prickly sida plants. After the 72 h AAP, 20 viruliferous or non-viruliferous whiteflies were tested for the presence of SiGMV via PCR using primers SiGMVF and SiGMVR. Total DNA was extracted from individual whiteflies using a specially formulated Chelex resin, InstaGene matrix (Bio-Rad, Hercules, CA), following manufacturer’s instructions and subjected to PCR analysis. Twenty plants of the same size from each test plant species were placed in clean whitefly-proof cages at 10 plants per cage (2 cages for each plant species). Using a clip cage, 100 viruliferous (from a batch that tested at least 80% positive via PCR) or non-viruliferous whiteflies were attached to the first true leaf of every test plant placed within a cage. Cages were arranged in a completely randomized design on a greenhouse bench (10 × 1 m) and maintained under the conditions described above. Symptom development in plants was recorded for the next eight weeks. After four weeks of inoculation, 100 mg of tissue from the youngest leaf of the test plants was collected and surface sterilized (Appendix A). Using the GeneJET Plant Genomic DNA Purification Kit, the total DNA from surface-sterilized tissues was extracted. Water from the last rinsate (4 μL) and extracted total genomic DNA (20 ng) was subjected to PCR analysis using primers SiGMVF and SiGMVR (Appendix A). The percentage of SiGMV infection was measured. Plants of each species infested with non-viruliferous whiteflies served as negative control, and the experiment was conducted three times (*n* = 30 for each plant species).

Virus accumulation in the infected plants was estimated through quantitative PCR (qPCR). Primers designed inhouse, SiGMV-QF and SiGMV-QR (Appendix A) that amplified a 114 bp region of AV1 gene of SiGMV, were used for qPCR. Quantitative PCR was performed using GoTaq qPCR Master Mix (Promega, Madison, WI). The Master Mix (2X) was combined with forward and reverse primers (SiGMV-QF and SiGMV-QR) (final concentration of 0.25 μM), 20 ng of DNA, and nuclease-free distilled water for a final volume of 12.5 μL. Quantitative PCR was performed using a Mastercycler ep realplex^4^ (Eppendorf, Hauppauge, NY, USA). An initial denaturation step (2 min at 95 °C) was followed by 40 cycles of 95 °C for 15 s, 63 °C for 10 s, and 72 °C for 20 s. Melting curve analysis was performed to test the specificity of binding. Absolute virus copy numbers were estimated using the procedure described by Legarrea et al. 2015 [40].

### 2.5. Virus Accumulation in Whiteflies Feeding on SiGMV-Infected Plants

Single SiGMV-infected or non-infected plants (status confirmed via PCR) from seven plant species (prickly sida, hollyhock, marsh mallow, country mallow, okra, snap bean, and tobacco) were placed in clean whitefly-proof cages. Cages were placed in a completely randomized design on a greenhouse bench (10 × 1 m) and maintained under the conditions described above. Whiteflies of up to 48 hours old that developed on cotton were collected using an aspirator and attached to the leaves of an infected or non-infected plant using clip cages (100 adults/cage/plant species). After a 72 h AAP, in order to remove virus (particles/DNA) from midgut lumen and hindgut, whiteflies in clip cages were re-attached to four-week-old cotton plants maintained in the greenhouse in whitefly-proof cages under the conditions described above. Forty-eight hours later, twenty whiteflies (corresponding to each infected or non-infected host) were randomly collected and placed in vials with 70% ethanol. Whiteflies were then stored at −80 °C until DNA extraction. In order to remove any residual virus present in honeydew on whiteflies, prior to DNA extraction, whiteflies were surface sterilized (Appendix A). Total DNA from individual whiteflies was extracted and subjected to PCR as described above. Percentages of whiteflies that acquired SiGMV from each acquisition host were measured. SiGMV accumulation in individual whiteflies was estimated through qPCR as described above. Twenty insects were used for every SiGMV-infected or non-infected host plant species. The experiment was conducted three times (*n* = 60).

### 2.6. Back Transmission of SiGMV from Different Host Plants to Snap Bean

SiGMV-infected or non-infected plants obtained in the previous experiment served as inoculum sources, and snap bean plants at the two-true-leaf stage (~3 weeks old) served as recipients. Using a clip cage, viruliferous whiteflies obtained as per the protocol described above were attached to snap bean plants (100/plant). Plants with clip cages were placed in insect proof cages, and after a one-week IAP, plants were sprayed with imidacloprid (Admire Pro, Bayer CropScience LP, NC). Plants were then maintained in the greenhouse for three weeks, after which 100 mg of topmost leaf sample was excised and surface-sterilized using a six-step surface sterilization protocol (Appendix A). Total genomic DNA from surface-sterilized leaf tissues was extracted with the GeneJET Plant Genomic Purification Kit. Virus infection status and quantity in plants was determined through end-point PCR and qPCR, respectively, as described above. Each treatment had 5 replications and the experiment was conducted three times (*n* = 15).

### 2.7. Whitefly Survival, Development, and Fecundity on SiGMV-Infected and/or Non-Infected Host Plants

Whitefly fitness parameters (developmental time, egg-to-adult survival, and fecundity) were evaluated on SiGMV-infected vs. non-infected host plants of each species. Using a clip cage, a pair of whitefly adults (~24 h old) were attached to infected or non-infected plants of each species for 48 h. After 48 h IAP, whiteflies were removed, and number of eggs laid were enumerated. Under the dissecting microscope, using a micro-needle, the number of eggs on each leaf was adjusted to three. Simultaneously, a leaf figure along with the egg’s position on the leaf was drawn on an A4 Sheet. For the next five weeks, eggs were observed every morning (9–11 AM), and the egg-to-adult developmental time and survival percentage were recorded. Each treatment had five replications and the experiment was conducted three times (*n* = 45 for each plant species). Fecundity data were collected for two weeks post adult emergence. Using clip cages, pair of whitefly adults (~48 h old) were attached to the leaves of infected or non-infected host plant of each species of the same age. Whitefly adults were transferred to new clip cages every fifth day for the next 15 days. Numbers of eggs on leaves were recorded under the dissecting microscope (10X). Each treatment had 10 replications and the experiment was conducted three times (*n* = 30 for each plant species).

### 2.8. Statistical Analyses

Data from experimental repeats were pooled and data were analyzed in R version 3.6.0 [41]. Differences in the percent infection (infected vs. non-infected) in whiteflies and plants and whitefly survival were evaluated using binary logistic regression. Logistic model was fitted using *glmer* function by setting the family argument as binomial. Treatments were considered as fixed effects, and the experiment repeats and replications were considered as random effects. Analysis of variance (ANOVA) was run on the model using function *Anova* in the car package [42]. Post hoc test was performed with *glht* function using Tukey adjustments for pairwise comparisons in the multcomp package [43]. Data for SiGMV accumulation in whiteflies and plants and fecundity were analyzed with the Lme4 package using a linear mixed model [44]. Virus accumulation and fecundity data were log-transformed prior to analysis. During analysis, replications and repeats were considered as random effects and treatments were considered as fixed effects. ANOVA was run on the model using function *Anova* in the car package [42]. Means for virus accumulation were compared by Tukey post hoc tests with the *contrast* and *lsmeans* functions from the *lsmeans* package [45]. The median development time from egg to adult was analyzed by nonparametric Kruskal–Wallis and Wilcoxon rank-sum test (Mann–Whitney U test). Statistical differences were considered significant at *p* < 0.05.

### 2.9. Nucleotide Similarity and Phylogenetic Analysis

#### 2.9.1. SiGMV DNA A Sequencing for Phylogenetic Analyses

Total DNA obtained from infected prickly sida was used to obtain the complete DNA-A sequence of SiGMV. SiGMVf 5’-CCTAAGCGCGATTTGCCAT-3’ and SiGMVr 5’-TACAGGGAGCTAAATCCAGCT-3’ primers specific to SiGMV were used to amplify the 1.5 kb region of the DNA-A [31]. The remaining portion of DNA-A was amplified through the degenerate primer pairs PAR1c496 and PAL1v1978 [34]. Amplicons obtained from specific and degenerate primer pairs were purified and sequenced. Obtained sequences were compiled to generate the complete sequence of DNA-A (2645 bp, GenBank Accession No. MK387701). Sequenced DNA-A shared 97% identity with SiGMV (GenBank Accession No. GQ357649) isolated from *P. vulgaris* from Alachua County Florida in 2006 [31].

#### 2.9.2. Phylogenetic Analysis

Single complete DNA-A sequence obtained in this study from a Georgia isolate, along with DNA-A sequences of sida viruses downloaded from GenBank, were used for phylogenetic analysis (Appendix A). Phylogenetic analysis was performed with the R statistical program version 3.6.0 [41]. All sequences were aligned with the MSA package [46]. For phylogenetic analysis, the function *modelTest* was used to compare different nucleotide substitution models [47]. The best fitting model (general time-reversible model) was selected based on the Akaike Information Criterion (AIC). A phylogenetic tree was constructed using the maximum likelihood (ML) method using the *optim.pml* command in the Phangorn package [47]. Using the *bootstrap.pml* command in the Phangorn package, bootstrap support values were assigned to the nodes. A phylogenetic tree obtained in the Newick format from R was visualized in the Interactive Tree of Life, version 4 [48]. Partial sequences from five 2019 Florida isolates were also compared with Georgia and GenBank sequences.

## 3. Results

### 3.1. SiGMV Symptoms and Accumulation in Inoculated Host Plants

SiGMV infection between test plant species was significantly different (*χ*2_7, 232_ = 23.41; *p* < 0.01). Among the eight plant species tested, except cotton, all were infected with SiGMV. The highest percent infection was detected in hollyhock and pricky sida, followed by country mallow, snap bean, marsh mallow, tobacco, and okra plants (Figure 2A). Typical SiGMV infection symptoms were observed after three–five weeks post-inoculation (Figure 3). Severe golden mosaic was observed on hollyhock, country mallow, and prickly sida plants. Symptoms on snap bean and marsh mallow included chlorotic spots, leaf crumpling, mild foliar golden mosaic and stunted growth. Young leaves of okra plants developed mild chlorotic spots. Symptoms on tobacco plants included thickened and leathery young leaves (Figure 3). In addition, SiGMV accumulation differed significantly between plant species (*F*_6, 163_ = 14.58; *p* < 0.01). Highest SiGMV accumulation was observed in leaf tissues of hollyhock and prickly sida followed by snap bean, marsh mallow, and okra (Figure 2B). SiGMV accumulation was lowest in country mallow, and it did not differ from tobacco plants (Figure 2B). None of the plants subjected to non-viruliferous whiteflies feeding developed SiGMV-associated symptoms or whitefly feeding-associated physiological disorders and were negative for SiGMV following endpoint PCR analysis.

### 3.2. SiGMV Detection and Accumulation in Whiteflies

SiGMV DNA-A was detected in *B. tabaci* adults that were subjected to a 72 h AAP on infected host plants, verifying that whiteflies were able to successfully acquire virus from all infected plant species. SiGMV acquisition by whiteflies varied between host plants (*χ*2_7, 413_ = 83.39; *p* < 0.01). The highest acquisition was observed in whiteflies feeding on prickly sida and hollyhock plants, followed by snap bean and tobacco plants (Figure 4A). Similar percentages of SiGMV acquisition were observed in whiteflies feeding on okra and marsh mallow plants. The lowest percentage of acquisition was observed in whiteflies feeding on country mallow plants (Figure 4A). Trends of SiGMV accumulation in whiteflies were similar to virus accumulation in host plants (Figure 4B). Whiteflies feeding on prickly sida plants accumulated significantly higher amounts of SiGMV than whiteflies feeding on all other plants (*F*_6, 206_ = 18.9; *p* < 0.01). No differences were observed in the virus accumulation in whiteflies feeding on snap bean and marsh mallow plants (Figure 4B). Similarly, no difference was observed with whiteflies feeding on tobacco and okra plants. Lowest accumulation was observed in whiteflies feeding on country mallow plants.

### 3.3. Back Transmission of SiGMV from Different Host Plants to Snap Bean Plants

Percent SiGMV infection in snap bean plants was dependent upon the SiGMV inoculum source (*χ*2_7, 98_ = 69.39; *p* < 0.01). After a 72 h AAP on SiGMV-infected source plants, whiteflies were able to acquire SiGMV from all infected plants. However, whiteflies provided with an AAP on snap bean, hollyhock, prickly sida, marsh mallow, and okra alone were able to successfully inoculate recipient snap bean plants. Highest SiGMV infection percentage in snap bean plants occurred when prickly sida and hollyhock plants were the inoculum sources (Figure 5A), while lowest infection percentage was observed with okra plants as inoculum sources. Likewise, SiGMV accumulation in infected snap bean plants was inoculum source-dependent (*F*_5, 70_ = 27.3; *p* < 0.01). Highest SiGMV accumulation was recorded in snap bean plants inoculated by whiteflies that acquired the virus from prickly sida and hollyhock plants (Figure 5B), while lowest SiGMV accumulation was detected in the snap bean inoculated by whiteflies that acquired SiGMV from infected okra plants. SiGMV infection symptoms on recipient snap bean plants varied with inoculum sources and developed between three and five weeks post-inoculation (Figure 6). Severe leaf crumpling was observed in infected snap bean plants when prickly sida, hollyhock, and snap bean were used as inoculum sources. Mild crumpling was observed on snap bean plants when marsh mallow and okra plants served as inoculum sources. No snap bean plant infection was recorded when tobacco or country mallow plants were used as inoculum sources.

### 3.4. Whitefly Survival, Development, and Fecundity on Infected and/or Non-Infected Host Plant Species

*B. tabaci* B developmental parameters between non-infected plant species varied significantly (Table 1). Percent survival from egg to adult differed significantly between non-infected host plant species (Table 1). Significantly higher egg-to-adult survival rate was recorded on hollyhock compared to other plant species. The lowest survival was recorded on tobacco plants, and it was comparable to survival on okra, marsh mallow, country-mallow, and prickly sida plants. Similarly, egg-to-adult developmental time and fecundity varied significantly between non-infected plant species (Table 1). Whitefly egg-to-adult developmental time on country mallow and prickly sida plants was significantly higher than developmental time on snap bean, hollyhock, and marsh mallow plants and the lowest developmental time was recorded on marsh mallow plants. Whiteflies laid significantly more eggs on okra and snap bean plants compared with other host plants. Furthermore, mean number of eggs laid on marsh mallow plants was significantly higher than that on tobacco, country-mallow, and prickly sida plants.

Whiteflies were able to complete their life cycle on all tested non-infected and infected plant species (Table 2). The survival from egg to adult did not differ between infected and non-infected tobacco, snap bean, hollyhock, country mallow, marsh mallow, prickly sida, or okra plants (Table 2). Similarly, whitefly median developmental time from egg to adult did not differ between infected and non-infected tobacco, snap bean, hollyhock, or marshmallow (Table 2). However, median developmental time from egg to adult of infected and non-infected prickly sida, country mallow, or okra were significantly different from each other (Table 2). Furthermore, mean number of eggs laid by whiteflies did not differ between infected and non-infected tobacco, snap bean, hollyhock, or marsh mallow (Table 2). However, mean number of eggs laid by whiteflies differed significantly between whiteflies feeding on SiGMV-infected and non-infected prickly sida, country mallow, or okra (Table 2). Whiteflies required significantly less time to develop and laid significantly more eggs on infected prickly sida and country mallow in comparison with non-infected prickly sida and country mallow plants, respectively. In contrast, whiteflies required significantly less time to develop and laid significantly more eggs on non-infected okra in comparison with SiGMV-infected okra plants.

### 3.5. Nucleotide Similarity and Phylogenetic Analysis

BLASTn analysis of DNA-A sequence of the Georgia isolate used in this study revealed a 97% nucleotide identity with the sequence of SiGMV isolate (GenBank Accession No. GQ357649) previously reported from Alachua County, Florida, in 2006 [31]. Old and New World sida virus DNA-A sequences from the GenBank and this study are represented in the phylogenetic tree (Figure 7). Sida viruses reported from the New World formed multiple paraphyletic groups. Furthermore, two NW viruses (SiMBV2 and SiGYSV) were closely related to the OW viruses and formed a monophyletic clade with them (Figure 7). In addition, the SiGMV reported in the current study and previously from Florida were closely related with sida yellow mottle virus (SiYMoV) detected in Cuba. All 2019 Florida samples were determined to be infected with SiGMV isolates and shared 93–99% nucleotide sequence identity with the Georgia isolate used in this study and with the previously reported Florida isolate. In addition, one plant from St. Lucie County, Florida was determined to contain a mixed infection with macroptilium yellow mosaic Florida virus, yet another begomovirus.

## 4. Discussion

SiGMV is emerging as a major issue in snap bean production systems in the southeastern United States, particularly in Georgia, and it is also being noted in Florida. Several ecological factors such as host range, symptom expression, and vector-mediated transmission along with virus genetics can play crucial roles in the emergence and development of virus epidemics. The current study examined the experimental host range of SiGMV in the greenhouse. Of the eight hosts evaluated, snap bean, tobacco, hollyhock, country mallow, marsh mallow, prickly sida, and okra were determined to be susceptible to SiGMV. Cotton was not a SiGMV host. Follow-up transmission studies revealed that whiteflies were able to transmit SiGMV from several infected hosts to snap bean. Whitefly fitness studies revealed that *B. tabaci* B was able to complete its life cycle on all infected plant species. Furthermore, SiGMV infection in prickly sida and country mallow plants positively influenced whitely development and reproduction. Among evaluated plants, mallow species such as prickly sida seemed to be of relevance for SiGMV epidemics, as this mallow weed could potentially overwinter in Georgia and Florida, and can function as virus inoculum sources and as vector reservoirs. Phylogenetic analysis revealed that SiGMV identified in this study is more closely related to the sida viruses reported from the New World (NW) compared with sida viruses reported from the Old World (OW).

The experimental host range study shows that SiGMV can infect plants belonging to three different families, including commercially important crops such as okra, snap bean, and tobacco. Since this study used vector-mediated transmission in place of plasmid mediated transmission, the results here depict more realistic scenarios that might be observed under field conditions. This study also observed a direct relationship between severity of symptoms and the level of SiGMV accumulation in susceptible host plants (hollyhock, prickly sida, snap bean, marsh mallow, country mallow, tobacco, and okra). Earlier studies with tomato yellow leaf curl virus (TYLCV, begomovirus) have shown that whiteflies strongly prefer susceptible tomato genotypes with severe disease phenotypes and higher virus accumulation levels [40]. Such preference could play a crucial role in SiGMV epidemics. However, at this time, it is unclear how variations in disease phenotype in different hosts will affect the vector preferences and SiGMV epidemics. Further studies are warranted on the effects of SiGMV disease phenotype on vector feeding preference.

Except for prickly sida and marsh mallow, a direct relationship between SiGMV accumulation in whiteflies and SiGMV accumulation in susceptible host plants was observed. Although prickly sida and hollyhock plants accumulated the same amount of SiGMV DNA, whiteflies feeding on prickly sida acquired more viral DNA and more virions. Snap bean plants accumulated significantly more SiGMV than marsh mallow plants did. However, whiteflies feeding on snap bean and marsh mallow accumulated similar amounts of SiGMV. Previous studies with electrical penetration graphs have shown that whiteflies spending more time feeding acquire more virus than whiteflies with interrupted feeding [49]. One possible explanation is that whiteflies might be spending more time feeding on prickly sida and marsh mallow than on other hosts, resulting in increased opportunities for virus acquisition. Whiteflies were able to acquire virus from country mallow plants with the lowest SiGMV accumulation, confirming that sufficient SiGMV was present in this host for acquisition by whiteflies. In the subsequent back transmission study, it was evident that percent infection, SiGMV accumulation, and disease symptoms in recipient snap bean plants were dependent upon SiGMV accumulation in source plants and whiteflies—a density-dependent phenomenon. This observed phenomenon lends itself to a speculation as to why whiteflies feeding on SiGMV-infected tobacco and country mallow plants acquired SiGMV but failed to successfully inoculate SiGMV to the recipient snap bean plants. The number of virus particles inoculated by whiteflies might have been below the inoculation threshold required to establish infection in the snap bean cultivar used in this study. Earlier studies also have reported a direct correlation between TYLCV accumulation in whiteflies following feeding on source plants with varying levels of TYLCV and subsequent disease severity in susceptible plants [40,50,51]. Results from the present study and earlier studies show that the higher the begomovirus accumulation level in the source plant, the higher the begomovirus accumulation in the whitefly, and more severe the disease phenotype in the recipient susceptible plants. However, it may be premature to assume such a direct relationship based on research with two begomoviruses alone.

Whiteflies were able to complete their life cycle on SiGMV-susceptible plant species indicating that all tested plant species could serve as reservoirs for whiteflies. SiGMV modulated effects on *B. tabaci* B were context-specific and dependent upon the SiGMV–host combination. Previous studies with other begomoviruses have reported similar results [52,53]. Therefore, it is very likely that a given begomovirus can modulate multiple-host plants (species) differently, resulting in a spectrum of interactions (negative–neutral–fit) between infected host plants and vectors. Mechanisms behind these varied interactions are not fully understood and warrant future investigation.

*Sida* spp. belong to the mallow family, Malvaceae. Mallow plants are widely distributed throughout the Americas [54], and as reported here, SiGMV can infect multiple mallow species. Mallows such as prickly sida are perennial weeds in several crops and in the landscape in the southeastern United States and can serve as inoculum sources for SiGMV [55]. Prickly sida accumulates sufficient levels of SiGMV, and upon SiGMV infection, performance of *B. tabaci* B is significantly improved on prickly sida plants. Furthermore, *B. tabaci* B was able to transmit SiGMV to snap bean after a 72 h AAP on infected prickly sida plants. Along with serving as a source of primary inoculum, over the long term, prickly sida can assist in increasing the diversity of SiGMV through recombination and/or other genetic mechanisms [56,57]. Earlier studies have reported that once a begomovirus becomes established in commercially cultivated crops, it quickly adapts itself and evolves to become distinct from parental populations, thereby reaffirming their quasispecies status [8]. Therefore, whiteflies transmitting begomoviruses between cultivated and non-cultivated hosts can contribute to SiGMV evolution and disease epidemics [58,59]. Taken together, prominent and/or prevalent hosts such as prickly sida can play crucial roles in emergence of SiGMV epidemics in multiple susceptible crops such as okra and snap bean.

Phylogenetic relationships of sida viruses with other related begomoviruses have been previously studied [60]. Begomoviruses are considered separate species when there is a <91% pairwise identity in the DNA-A of different viruses, and the cutoff for strain demarcation is set at 94% [61]. This study reports phylogenetic relationships between sida viruses identified in different parts of the world. Based on nucleotide sequence analysis, the Georgia SiGMV isolate reported here is identical to the Florida SiGMV isolate collected in 2006 [31]. The results of this study indicate that SiGMV is more closely related to sida viruses reported from the New World, particularly to Sida yellow mottle virus (SiYMoV), sida mosaic Sinaloa virus (SiMSiV), and sida ciliaris golden mosaic virus (SicGMV) reported from Cuba, Mexico, and Venezuela, respectively.

## 5. Conclusions

Epidemics of plant viruses are dependent on the availability of vector reservoirs and virus inoculum sources that favor development of explosive populations of the vector as well as rapid acquisition and inoculation of viruses [62]. As reported here, SiGMV can infect multiple weeds/crop plants and reproduce efficiently on them. Whiteflies can acquire SiGMV from these plants and successfully inoculate the virus to susceptible plants. Moreover, weeds and crop hosts identified as inoculum sources and vector reservoirs are widespread in Georgia and in Florida [55]. Therefore, it is possible that SiGMV has established itself in the southeastern US since its first detection in Florida and Georgia, and may continue to emerge as a major threat to snap bean production and possibly other crops. Additionally, SiGMV has often been detected as a mixed infection with CuLCrV in snap bean plants in Georgia [29,30] and once with macroptilium yellow mosaic Florida virus. Mixed infection of plant viruses is ubiquitous in nature [63,64,65], and viruses involved in mixed infection could interact synergistically [66]. Several agriculturally important plant viruses produce enhanced disease phenotypes in mixed infection status [67,68,69]. It is possible that mixed infection of SiGMV with other plant viruses could trigger severe epidemics in the future. Therefore, understanding the interactions of SiGMV with other established viruses in the region will help better evaluate the risk posed by SiGMV.

## Figures and Tables

**Figure 1 viruses-15-00357-f001:**
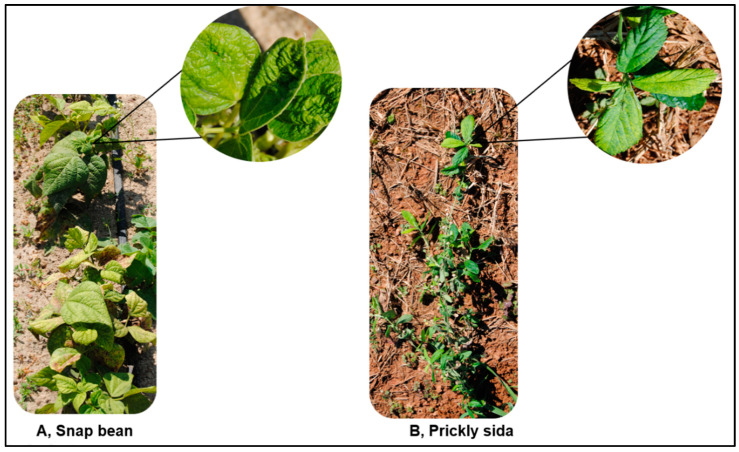
Snap bean and prickly sida plants infected with SiGMV in a snap bean field in Tifton, GA, USA.

**Figure 2 viruses-15-00357-f002:**
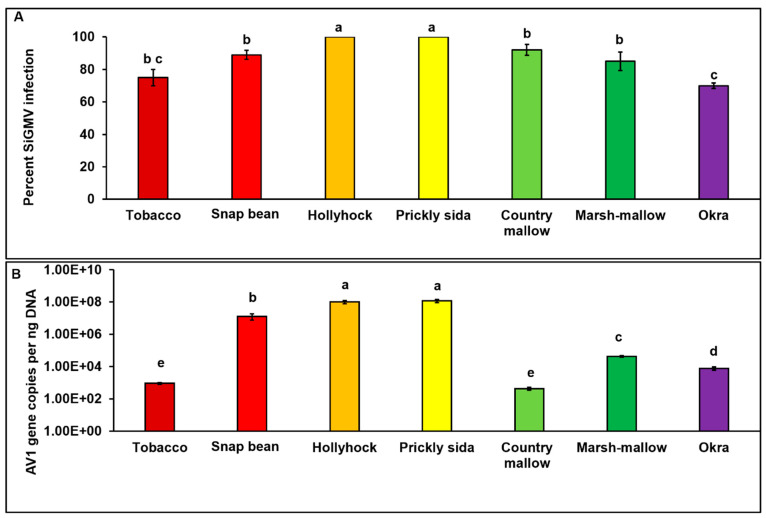
**Percent SiGMV infection and accumulation in different host plants**. (**A**) Bars with standard errors represent the average percent infection in susceptible plant species. (**B**) Bars with standard errors represent the average number of SiGMV AV1 copies accumulated in the leaf tissues of SiGMV-infected plant species. AV1 gene copy numbers were estimated by qPCR followed by absolute quantitation using plasmids containing SiGMV AV1 gene inserts as standards. Different letters on bars indicate significant differences between means at *α* = 0.05. For copy numbers, Y-axis is on logarithmic scale.

**Figure 3 viruses-15-00357-f003:**
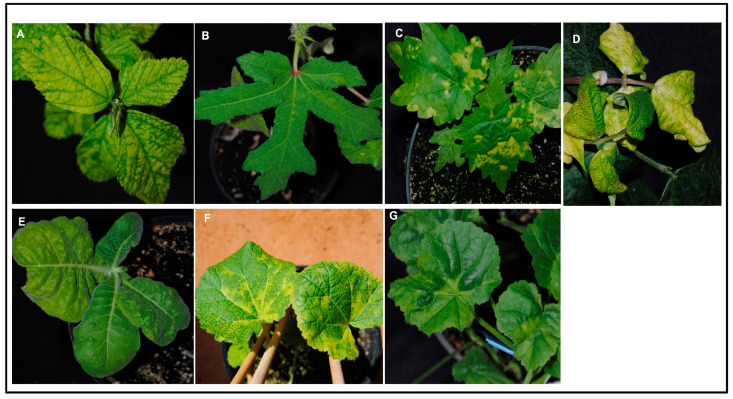
**Symptoms on SiGMV -infected plants following inoculation by viruliferous whiteflies.** Viruliferous whiteflies were generated by providing a 72h AAP on SiGMV-infected prickly sida plants. (**A**) Prickly sida; (**B**) okra; (**C**) country mallow; (**D**) snap bean; (**E**) tobacco; (**F**) hollyhock; and (**G**) marsh mallow. Photographs were taken at the most symptomatic phase of SiGMV infection.

**Figure 4 viruses-15-00357-f004:**
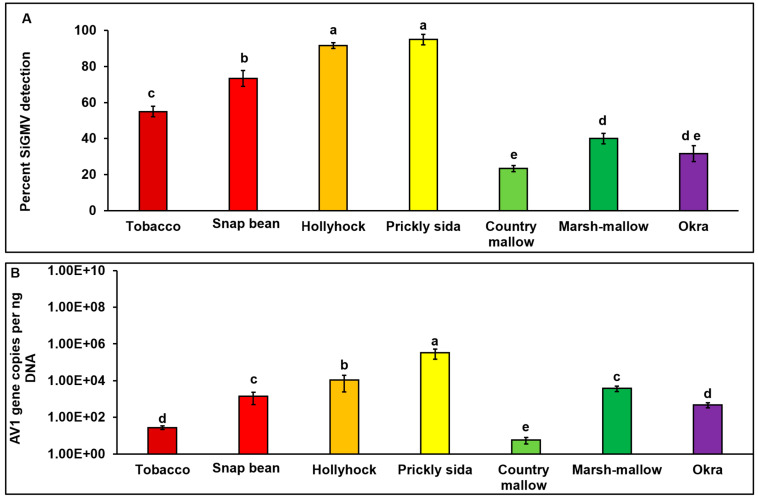
**Percent infection and SiGMV accumulation in whiteflies feeding on infected plants.** (**A**) Bars with standard errors represent the average percent SiGMV detection in whiteflies feeding on SiGMV-infected plant species. (**B**) Bars with standard errors represent the average number of SiGMV AV1 gene copies accumulated in whiteflies feeding on SiGMV-infected plants of various species. AV1 gene copy numbers for SiGMV were estimated by qPCR followed by absolute quantitation using plasmids containing SiGMV AV1 gene inserts as standards. Different letters on bars indicate significant differences between means at *α* = 0.05. For copy numbers, Y-axis is on logarithmic scale.

**Figure 5 viruses-15-00357-f005:**
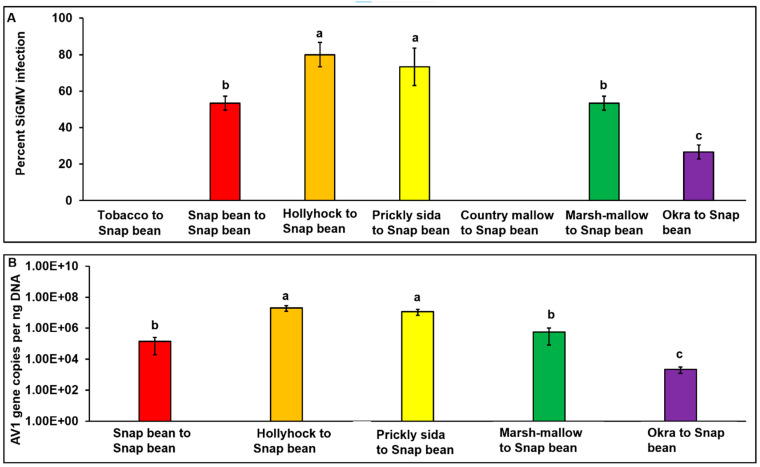
**Percent infection and SiGMV accumulation in recipient snap bean plants.** (**A**) Bars with standard errors represent the average percent SiGMV infection in snap bean plants inoculated by SiGMV viruliferous whiteflies after a 72 h AAP on SiGMV-infected tobacco, snap bean, hollyhock, prickly sida, country mallow, marsh mallow, and okra plants. (**B**) Bars with standard errors represent the average number of SiGMV copies accumulated in infected snap bean plants after inoculation by whiteflies that acquired SiGMV from SiGMV-infected snap bean, hollyhock, prickly sida, marsh mallow, and okra plants. Copy numbers of SiGMV AV1 gene were estimated by qPCR followed by absolute quantitation using plasmids containing SiGMV AV1 gene inserts as standards. Different letters on bars indicate significant differences between means at *α* = 0.05. For copy numbers, Y-axis is on logarithmic scale.

**Figure 6 viruses-15-00357-f006:**
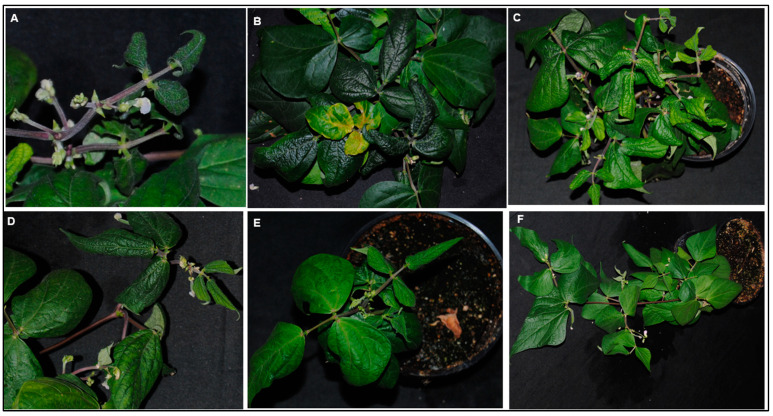
Symptoms on snap bean plants upon infection with SiGMV after inoculation by whiteflies that had acquired SiGMV from different SiGMV-infected host plants. Snap bean infected by whiteflies after 72 h AAP on (**A**) SiGMV-infected snap bean; (**B**) SiGMV-infected prickly sida; (**C**) SiGMV-infected hollyhock; (**D**) SiGMV-infected marsh mallow; and (**E**) SiGMV-infected okra. (**F**) Non-infected snap bean plant subjected to non-viruliferous whiteflies feeding. Symptoms were observed three to four weeks post-inoculation. Photographs were taken at the most symptomatic phase of SiGMV infection.

**Figure 7 viruses-15-00357-f007:**
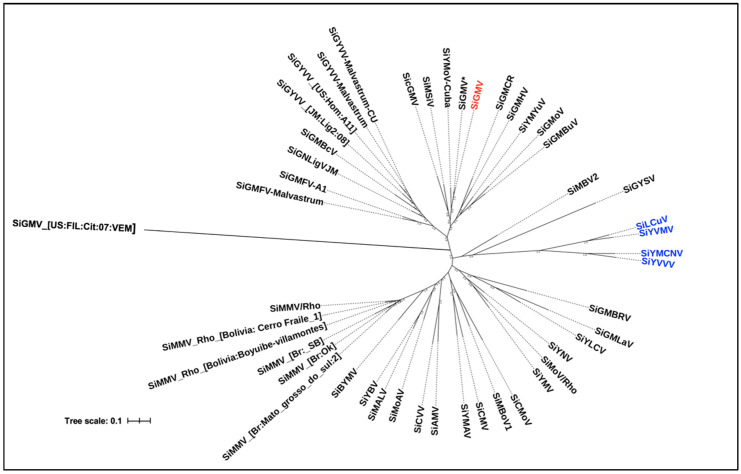
Phylogenetic tree illustrating the relationships among the DNA-A of SiGMV isolate used in this study and previously described sida viruses. Maximum Likelihood (ML) phylogenetic tree was constructed based on the alignment of the full-length DNA-A sequences of sida viruses obtained from GenBank and DNA-A sequence of SiGMV reported in this study. The best fitting model was selected based on the Akaike Information Criterion (AIC). Phylogenetic tree was constructed using *optim.pml* command in Phangorn package in R. Support for nodes in a bootstrap analysis with 1000 replicates, which is shown. Tree branch lengths are drawn to scale and bar indicates number of nucleotide differences between sequences. Node tip shows virus abbreviations according to the International Committee on Taxonomy of Viruses. The Georgia SiGMV isolate reported in this study is shown in red. SiGMV reported in 2006 from Florida is highlighted with an asterisk, and the Old World and New World sida viruses are shown in blue and black, respectively.

**Table 1 viruses-15-00357-t001:** Survival, developmental time, and fecundity of *B. tabaci* B on non-infected plant species.

Plant Species	Treatment	N ^a^	Egg-to-Adult Survival ^b^		Egg-to-Adult Developmental Time ^c^		N ^d^	Fecundity ^e^	
Tobacco	Non-infected	45	46.22± 11.75 c	*χ*2_6, 308_ = 31.56; *p* < 0.001	24 (22–35) b	*χ*2_6, 195_ = 63.03; *p* < 0.001	30	77.5 ± 15.37 c	*F*_6, 203_ = 19.39; *p* < 0.001
Snap bean	Non-infected	45	62.33 ± 2.22 b		22 (18–26) c		30	153.3 ± 10.81 a	
Hollyhock	Non-infected	45	86.67 ± 3.84 a		22 (19–36) c		30	108.3 ± 12.10 b c	
Prickly sida	Non-infected	45	58.12 ± 9.23 b c		26 (20–34) a b		30	72.67 ± 9.67 c	
Country mallow	Non-infected	45	50 ± 10.18 b c		26 (24–31) a b		30	77.45 ± 7.58 c	
Marsh mallow	Non-infected	45	54.65 ± 9.21 b c		20 (21–37) d		30	112.04 ± 10.38 b	
Okra	Non-infected	45	56.23 ± 13.22 b c		23 (21–33) c b		30	162.34 ± 11.75 a	

SiGMV—sida golden mosaic virus. ^a^ Number of eggs monitored to adulthood. ^b^ Mean survival from egg to adult on non-infected and SiGMV-infected plant species. ^c^ Median developmental time (days) from egg to adult with range in parentheses. ^d^ Number of whitefly pairs. ^e^ Fecundity (mean number of eggs ± SE) of whiteflies on non-infected plant species for 15 days.

**Table 2 viruses-15-00357-t002:** Survival, developmental time, and fecundity of *B. tabaci* B on non-infected and SiGMV-infected plant species.

Plant Species	Treatment	N ^a^	Egg-to-Adult Survival ^b^		Egg-to-Adult Developmental Time ^c^		N ^d^	Fecundity ^e^	
Tobacco	Non-infected	45	46.22 ± 11.75	*χ*2_1.88_ = 2.28; *p* = 0.13	24 (22–35)	*U*_1.46_ = 32, *p* = 0.17	30	77.5 ± 15.37	*F*_1.58_= −1.44; *p* = 0.07
	Infected	45	57.77 ± 5.87		25 (19–34)		30	87.5 ± 22.1	
Snap bean	Non-infected	45	62.33 ± 2.22	*χ*2_1.88_ = 0.63; *p* = 0.43	22 (18–26)	*U*_1.58_ = 73, *p* = 0.41	30	153.3 ± 10.8	*F*_1.58_ = 0.77; *p* = 0.22
	Infected	45	64.44 ± 8.01		22 (20–31)		30	158.3 ± 9.1	
Hollyhock	Non-infected	45	86.67 ± 3.84	*χ*2_1.88_ = 2.26; *p* = 0.42	22 (19–36)	*U*_1.72_ = 97, *p* = 0.74	30	108.3 ± 12.1	*F*_1.58_ = −0.15; *p* = 0.43
	Infected	45	80.00 ± 3.84		21 (20–29)		30	111.4 ± 6.57	
Prickly sida	Non-infected	45	58.12 ± 9.23	*χ*2_1.88_ = 2.25; *p*= 0.13	26 (20–34)	*U*_1.56_ = 42, *p* < 0.001	30	72.67 ± 9.67	*F*_1.58_ = −4.3; *p* < 0.001
	Infected	45	68.4 ± 10.11		20 (18–29)		30	112.23 ± 8.87	
Country mallow	Non-infected	45	50 ± 10.18	*χ*2_1.88_ = 1.61; *p* = 0.20	26 (24–31)	*U*_1.68_ = 13, *p* = 0.013	30	77.45 ± 7.58	*F*_1.58_ = −3.72; *p* < 0.001
	Infected	45	60.55 ± 8.01		22 (21–28)		30	101.60 ± 10.87	
Marsh mallow	Non-infected	45	54.65 ± 9.21	*χ*2_1.88_ = 0.84; *p* = 0.36	20 (18–37)	*U*_1.50_ = 82, *p* = 0.136	30	112.04 ± 10.38	*F* _1.58_= −1.09; *p* = 0.13
	Infected	45	60.25 ± 7.89		21 (17–31)		30	126.60 ± 9.87	
Okra	Non-infected	45	56.23 ± 13.22	*χ*2_1.48_ = 6.26; *p* = 0.09	23 (21–33)	*U*_1.88_ = 39, *p* < 0.001	30	162.34 ± 11.75	*F*_1.58_ = 5.81; *p* < 0.001
	Infected	45	51.11 ± 14.57		28 (22–32)		30	89.00 ± 22.1	

SiGMV—sida golden mosaic virus. ^a^ Number of eggs monitored to adulthood. ^b^ Mean survival from egg to adult on non-infected and SiGMV-infected plant species. ^c^ Median developmental time (days) from egg to adult with range in parentheses. ^d^ Number of whitefly pairs. ^e^ Fecundity (mean number of eggs ± SE) of whiteflies on non-infected or SiGMV-infected plant species for 15 days.

## Data Availability

Not applicable.

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
