# Peer review of "Sida Golden Mosaic Virus, an Emerging Pathogen of Snap Bean (Phaseolus vulgaris L.) in the Southeastern United States"

_viruses, 2023, doi:10.3390/v15020357_

Round 1

Reviewer 1 Report

This article: Sida golden mosaic virus, an emerging pathogen of snap bean (Phaseolus vulgaris L.) in the southeastern United States, presents novel information and deserve to be considered for publication. I have few points to make which may be addressed while preparing the revision.

Why was no snap bean infection recorded when tobacco or country mallow were used as inoculum sources. You should make a discussion in discussion.

Additional comments are embedded in the PDF file which may be taken care of while preparing the revision. Please, complete the work and submit a new manuscript.

Author Response

January 19, 2023

Dear Reviewer, 

My co-authors and I greatly appreciate your comments. We believe that your review and comments have improved our manuscript. We have carefully considered each comment and have tried to address them to the best of our abilities. Our explanation for each comment is included below. Our replies are in bold font.

This article: Sida golden mosaic virus, an emerging pathogen of snap bean (Phaseolus vulgarisL.) in the southeastern United States, presents novel information and deserve to be considered for publication. I have few points to make which may be addressed while preparing the revision.

Why was no snap bean infection recorded when tobacco or country mallow were used as inoculum sources. You should make a discussion in discussion. 

An explanation has been included in the discussion section in lines 531 through 540…

Whiteflies were able to acquire virus from country mallow plants with the lowest SiGMV accumulation, confirming that sufficient SiGMV was present in this host for acquisition by whiteflies. In the subsequent back transmission study, it was evident that percent infection, SiGMV accumulation, and disease symptoms in recipient snap bean plants were dependent upon SiGMV accumulation in source plants and whiteflies –a density dependent phenomenon. This observed phenomenon lends itself to a speculation as to why whiteflies feeding on SiGMV-infected tobacco and country mallow plants acquired SiGMV but failed to successfully inoculate SiGMV to the recipient snap bean plants. The number of virus particles inoculated by whiteflies might have been below the inoculation threshold required to establish infection in the snap bean cultivar used in this study”

Additional comments are embedded in the PDF file which may be taken care of while preparing the revision. Please, complete the work and submit a new manuscript.

Additional comments especially those pertaining to the methods section have been addressed by adding more sub-sections to enhance clarity and abbreviating sections. A few tables and one protocol are now included in the supplementary section to decrease the overall length. All other minor revisions pertaining to italicization and virus nomenclature in the reference section have been addressed and identified.

We appreciate the feedback. Thank you.

Yours sincerely,

Rajagopalbabu Srinivasan

Reviewer 2 Report

The authors provide a very nice manuscript on the detection of Sida golden mosaic virus in Georgia, exploring the virus’s host range, transmission, effects on vector fitness, and phylogenetic relationships with other Sida viruses. Few, mostly minor editorial corrections and suggestions are provided on the manuscript.

Some additional clarity is needed regarding data interpretation in Table 4/lines 444-448 (comments are provided on the manuscript) and on the phylogenetic analysis results/discussion (lines 474-476). I don’t follow your explanation of the tree topology, especially on the ‘two’ NW versus OW clades and the NW groups in lines 474-476. Clades indicate monophyly, and the NW viruses appear to be paraphyletic (or, as mentioned, in different groups). Further, two NW viruses (SiMBV2 and SiGYSV) appear to be more closely related to the OW viruses and form a monophyletic clade with them. Further discussion/explanation or, likely, correction is necessary here.

Overall, this is well-written, scientifically sound manuscript with important findings on an emerging virus.

Author Response

January 19, 2023

Dear Reviewer, 

My co-authors and I greatly appreciate your comments. We believe that your review and comments have improved our manuscript. We have carefully considered each comment and have tried to address them to the best of our abilities. Our explanation for each comment is included below. Our replies are in bold font.

The authors provide a very nice manuscript on the detection of Sida golden mosaic virus in Georgia, exploring the virus’s host range, transmission, effects on vector fitness, and phylogenetic relationships with other Sida viruses. Few, mostly minor editorial corrections and suggestions are provided on the manuscript.

Some additional clarity is needed regarding data interpretation in Table 4/lines 444-448 (comments are provided on the manuscript) and on the phylogenetic analysis results/discussion (lines 474-476). I don’t follow your explanation of the tree topology, especially on the ‘two’ NW versus OW clades and the NW groups in lines 474-476. Clades indicate monophyly, and the NW viruses appear to be paraphyletic (or, as mentioned, in different groups). Further, two NW viruses (SiMBV2 and SiGYSV) appear to be more closely related to the OW viruses and form a monophyletic clade with them. Further discussion/explanation or, likely, correction is necessary here.

This detail has been corrected and modified in the results section as suggested in lines 468-472…

Sida viruses reported from the New World formed multiple paraphyletic groups. Furthermore, two NW viruses (SiMBV2 and SiGYSV) were closely related to the OW viruses and formed a monophyletic clade with them (Fig. 7). Also, the SiGMV reported in the current study and previously from Florida were closely related with sida yellow mottle virus (SiYMoV) reported from Cuba.”

Overall, this is well-written, scientifically sound manuscript with important findings on an emerging virus.

We appreciate the feedback. Thank you.

All other minor comments raised on the manuscript also have been addressed and identified.

Yours sincerely,

Rajagopalbabu Srinivasan
